# The HadGEM3-GA7.1 radiative kernel: the importance of a well-resolved stratosphere

Christopher J. Smith[1,2], Ryan J. Kramer[3,4], and Adriana Sima[5]

[1]School of Earth and Environment, University of Leeds, Leeds LS2 9JT, United Kingdom
[2]International Institute for Applied Systems Analysis (IIASA), A-2361 Laxenburg, Austria
[3]Climate and Radiation Laboratory, NASA Goddard Space Flight Center, Greenbelt, MD 20771, USA
[4]Universities Space Research Association, 7178 Columbia Gateway Drive, Columbia, MD 21046, USA
[5]LMD/IPSL, Sorbonne Université, ENS, PSL Université, École polytechnique, Institut Polytechnique de Paris, CNRS, Paris France

**Correspondence:** C.J.Smith (c.j.smith1@leeds.ac.uk)

**Abstract.** We present top-of-atmosphere and surface radiative kernels based on the atmospheric component (GA7.1) of the HadGEM3 general circulation model developed by the UK Met Office. We show that the utility of radiative kernels for forcing adjustments in idealised $CO_2$ perturbation experiments is greatest where there is sufficiently high resolution in the stratosphere in both the target climate model and the radiative kernel. This is because stratospheric cooling to a $CO_2$ perturbation continues to increase with height, and low-resolution or low-top kernels or climate model output are unable to fully resolve the full stratospheric temperature adjustment. In the sixth phase of the Coupled Model Intercomparison Project (CMIP6), standard atmospheric model data is available up to 1 hPa on 19 pressure levels, which is a substantial advantage compared to CMIP5. We show in the IPSL-CM6A-LR model where a full set of climate diagnostics are available that the HadGEM3-GA7.1 kernel exhibits linear behaviour and the residual error term is small, and from a survey of kernels available in the literature that in general low-top radiative kernels underestimate the stratospheric temperature response. The HadGEM3-GA7.1 radiative kernels are available at https://doi.org/10.5281/zenodo.3594673 (Smith, 2019).

*Copyright statement.* TEXT

## 1 Introduction

Radiative kernels describe how a small change in an atmospheric state variable affects the Earth's energy balance (Soden et al., 2008; Shell et al., 2008). They allow an analysis of climate feedbacks (Shell et al., 2008; Soden et al., 2008; Sanderson and Shell, 2012; Jonko et al., 2012; Block and Mauritsen, 2013; Huang, 2013) or forcing adjustments (Vial et al., 2013; Zhang and Huang, 2014; Chung and Soden, 2015b; Smith et al., 2018; Myhre et al., 2018; Smith et al., 2020) from standardised climate model diagnostics such as those from the Coupled Model Intercomparison Projects (CMIPs). The use of radiative kernels is efficient, removing the need for time- and memory-consuming calculations of climate feedbacks online through partial radiative

perturbation calculations (Wetherald and Manabe, 1988) or offline using a standalone version of the model radiative transfer code (Colman and McAvaney, 2011).

A radiative kernel $K_X$ is in effect a four-dimensional (time, height, latitude, longitude) array describing how radiation fluxes $R$ change with an atmospheric state variable $X$

$$K_X(t,z,y,x) = \frac{\partial R}{\partial X}\bigg|_{(t,z,y,x)}. \tag{1}$$

Although strictly not a partial differential equation, this statement provides a concise written form and is used by others (e.g. Shell et al. (2008); Huang et al. (2017)). $R$ may be upwelling, downwelling or net, shortwave or longwave, radiation changes at any atmospheric level. Most commonly net top-of-atmosphere (TOA), surface and tropopause-level fluxes are of greatest interest. $X$ here represents atmospheric temperature ($T_a$), surface (skin) temperature ($T_s$), water vapour ($q$) and surface albedo ($\alpha$). Although other (non-cloud) variables may also be relevant, the majority of adjustments are expected to be captured under this framework (Vial et al., 2013). For determining adjustments to a radiative forcing $A_X$, the kernel $K_X$ is multiplied by the change in atmospheric state variable $\Delta X$ between two integrations of a climate model such that

$$A_X = K_X \Delta X. \tag{2}$$

$\Delta X$ is calculated as the difference of two atmosphere-only climate model integrations using climatological sea-surface temperatures and sea ice distributions, one of which is driven by a forcing perturbation (e.g. a quadrupling of $CO_2$) and the other a control. For temperature and albedo the adjustment is linear with $\Delta X$, and logarithmic for water vapour (Sanderson and Shell (2012) and Smith et al. (2018, Supplementary Material) describe how the adjustment to water vapour is applied in practice). For determining climate feedbacks $\lambda_X$, the perturbation is normalised by the change in global mean near-surface air temperature $T$ such that

$$\lambda_X = K_X \frac{\partial X}{\partial T}. \tag{3}$$

The individual contributions from each feedback component $\lambda_X$ contribute the total climate feedback $\lambda = \lambda_{T_a} + \lambda_{T_s} + \lambda_q + \lambda_\alpha + \lambda_c$ where $c$ represents cloud feedback in the forcing-feedback representation of the Earth's energy budget $\Delta N = F - \lambda \Delta T$. Here, $\Delta N$ is the Earth's energy imbalance and $F$ the effective radiative forcing. Likewise, the effective radiative forcing can be decomposed into

$$F = F_i + A_{T_a} + A_{T_s} + A_q + A_\alpha + A_c \tag{4}$$

with $F_i$ being the instantaneous radiative forcing (IRF).

Usage of radiative kernels assumes that radiative perturbations change linearly with changes in atmospheric state. Where perturbations are small, linearity is an appropriate assumption both for feedbacks (Jonko et al., 2012) and adjustments (Smith et al., 2018).

Cloud adjustments and feedbacks cannot be determined directly using atmospheric state kernels. They may be diagnosed using the cloud kernel based on International Satellite Cloud Climatology Project (ISCCP) simulator diagnostics (Zelinka et al.,

2012) or from the residual of all-sky and clear-sky radiative kernels (Soden et al., 2008; Shell et al., 2008). For adjustments this calculation is

$$A_c = (F - F^{\mathrm{clr}}) - (F_i - F_i^{\mathrm{clr}}) - \sum_{X \in \{T_a, T_s, q, \alpha\}} (A_X - A_X^{\mathrm{clr}}) \qquad (5)$$

where the clr superscript represents fluxes calculated in the absence of clouds. In Eq. (5), the instantaneous radiative forcing must be known or estimated. This method is commonly used, requiring the production of all-sky and clear-sky kernel sets to calculate all-sky and clear-sky adjustments.

This paper introduces the top-of-atmosphere and surface radiative kernels from the HadGEM3-GA7.1 model. HadGEM3-GA7.1 has 85 vertical levels up to 85 km vertical height (about 0.005 hPa). The construction of the HadGEM3-GA7.1 kernel was motivated by the observation that adjustments to a doubling of $CO_2$ in models participating in the Precipitation Driver and Response Model Intercomparison Project (PDRMIP; Myhre et al. (2017)) were around 0.3 W m$^{-2}$ larger using the ECMWF-Oslo kernel (Myhre et al., 2018) than other kernels used in the same study (Smith et al., 2018, Supplementary Figure 3). The ECMWF-Oslo kernel was built from ECMWF-Interim reanalysis data (Dee et al., 2011) which has 60 vertical levels up to 0.1 hPa. Most other kernels used in Smith et al. (2018) were derived from climate models with a lower model top, and it is possible that low-top kernels were underestimating the stratospheric adjustment in Smith et al. (2018).

For stratospheric temperature adjustments we compare the HadGEM3-GA7.1 kernel to other kernels in the literature using available $4\times CO_2$ results from climate models contributing to the Radiative Forcing Model Intercomparison Project (RFMIP). We find that in general, kernels based upon climate models with a high stratospheric resolution are better at resolving the stratospheric adjustment to a $CO_2$ forcing.

## 2 Construction of the HadGEM3-GA7.1 radiative kernel

One year of a pre-industrial, atmosphere-only (i.e. with climatological sea-surface temperatures and sea ice distributions) integration of the HadGEM3-GA7.1 general circulation model (Williams et al., 2018; Mulcahy et al., 2018) was run. HadGEM3-GA7.1 is the atmospheric component of the HadGEM3-GC3.1 physcial model and UKESM1.0 Earth System model that represents the UK research community's contribution to CMIP6. The model was run at LL (N96) resolution with a latitude-longitude grid of 1.25° by 1.875° and 85 vertical levels extending up to 85 km (approximately 0.005 hPa) and a native model timestep of 20 minutes. A pre-industrial base climatology for the kernels was chosen because the first identified use case for this kernel set was the RFMIP-ERF Tier 1 single-forcing experiments (Pincus et al., 2016). These experiments compare positive (e.g. greenhouse gas) and negative (e.g. aerosol) forcing perturbations with respect to a pre-industrial control baseline.

Model diagnostics of air temperature, specific humidity, surface (skin) temperature, surface albedo (ratio of broadband upwelling to downwelling shortwave surface radiation), model level pressure, surface pressure, cloud fraction, cloud water content, cloud ice content, effective (time-averaged) solar zenith angle and gridbox daylight fraction every two model hours were saved. 2-hourly sampling is considered to give an appropriate representation of the diurnal cycle of clouds and reduce biases due to variations in solar geometry with longitude compared to longer timesteps, while keeping computational demands

to a minimum. These model outputs were transplanted into an offline version of the SOCRATES radiative transfer code (version 17.03; Manners et al. (2015); Edwards and Slingo (1996)) and top-of-atmosphere and surface radiative fluxes calculated for each two-hour timestep in both the shortwave and longwave spectra, for all-sky and clear-sky. SOCRATES is a broadband radiation code that uses 6 bands in the shortwave and 9 bands in the longwave and is the same radiation scheme used in the HadGEM3-GC3.1 and UKESM1.0 climate models. Aerosols were neglected and greenhouse gases, including the prescribed CMIP6 monthly climatology in ozone concentrations, were set to their pre-industrial (1850) values. Following the protocol for RFMIP (Pincus et al., 2016), sea-surface temperatures and sea-ice distributions from 50 years of the HadGEM3-GC3.1 coupled model were used to build the climatology (Andrews et al., 2019). The kernel is therefore dependent on two aspects of the HadGEM3-GA7.1 model: the pre-industrial background climatology (including clouds), and the broadband version of the radiation code.

To build the kernel, each vertical level of the model on each 2-hour timestep was perturbed separately, firstly by 1 K for air temperature, and secondly by a perturbation in specific humidity that maintains relative humidity for an increase in 1 K (without actually changing the layer temperature). The surface temperature and surface albedo were also perturbed by 1 K and 1% (additive) individually each timestep. For each perturbation, surface and TOA fluxes are again saved for clear-sky and all-sky in the shortwave (SW) and longwave (LW), and the difference compared to the control simulation gives the radiative kernel for each model level or surface. Building the kernels took in total approximately three months of computing time on 24 processors on the University of Leeds "cluj" Linux cluster. Running the base HadGEM3 model at higher (MM) resolution, with five times as many grid points as LL, was not determined to be necessary, as kernels are usually ultimately regridded to the resolutions of other CMIP6 models which are approximately as coarse as LL (e.g. Table 1 in Smith et al. (2020)).

Following this, the air temperature and water vapour kernel outputs were normalised by multiplying by $10000\ \mathrm{Pa}/p_{\mathrm{thick}}$ where $p_{\mathrm{thick}}$ [Pa] is the thickness of each level in pressure co-ordinates. This allows the 85-level native model kernel to be reduced down to the 19-level standard CMIP6 pressure levels by providing a weighted average contribution to each pressure level. The kernels are further averaged by month. In the 19-level format they can be used with standard "Amon" model output from any CMIP6 model, which is one of the key advantages of radiative kernels.

## 3 Kernel profiles

### 3.1 Top-of-atmosphere kernels

Figure 1 shows the TOA radiative kernels for HadGEM3-GA7.1 for clear-sky and all-sky. The air temperature, all-sky kernel (Fig. 1a) shows a peak in cooling in the tropical upper troposphere, showing the importance of this region for changes in radiative balance. There are also substantial contributions to the TOA radiation balance from the lower troposphere in the mid-latitudes. For clear-sky (Fig. 1b) there is more latitude-height homogeneity in the troposphere, showing the impact of removing clouds. A key feature of the air temperature kernels is the increasing strength of the LW outgoing radiation with increasing stratospheric height. The temperature kernel is negative throughout the atmosphere, in keeping with the fact that an increase in temperature results in additional Planck emission of LW radiation to space.

Water vapour kernels (Fig. 1c,d) also show a peak in the upper tropical troposphere, which is opposite in sign to the negative temperature adjustment owing to the fact that water vapour is a significant greenhouse gas. In contrast to the temperature kernel, the water vapour kernel is very insensitive in the dry upper stratosphere.

The impact of cloud masking is more easily seen for the surface temperature kernels (Fig. 1e,f) and surface albedo kernels (Fig. 1g,h).

## 3.2 Surface kernels

Surface kernels are most useful for determining precipitation adjustments (Myhre et al., 2018) and feedbacks (Previdi, 2010), where the precipitation adjustment is proportional to the atmospheric absorption, calculated as the difference in TOA and surface adjustments. Figure 2 shows the surface radiative kernels for HadGEM3-GA7.1 for clear-sky and all-sky. Both the air temperature (Fig. 2a,b) and water vapour (Fig. 2c,d) kernels are more sensitive for perturbations close to the surface than higher in the atmosphere (note non-linear colour scales). Cloud masking for the surface temperature kernel has less of an effect for surface fluxes than for TOA fluxes (Fig. 2e,f), whereas the surface albedo kernel shows quite a similar spatial pattern (Fig. 2g,h) to its the TOA counterpart.

## 4 Comparison to other kernels for stratospheric temperature

Figure 3 shows the air temperature kernel for the stratosphere and upper troposphere for a selection of kernels available in the literature (Table 1). In all cases, radiative kernels have been interpolated from their native vertical resolution (except for CCSM4, which is available only on the standard 17 CMIP5 pressure levels) to the 19 CMIP6 pressure levels for consistency with CMIP6 model output. For our calculations of stratospheric temperature adjustment, where kernels do not extend up to the 1 hPa top level of CMIP6 model output, kernels have been extended upwards using the value from the highest level where data does exist, but in Fig. 3 missing data has been masked out. This extending upwards of the top level has been applied previously in adjustment calculations where the top level of the climate model is higher than the top level of the kernel (e.g. in Smith et al. (2018)). However, extending the top level of a radiative kernel upwards cannot make up for the fact that more radiation is emitted to space from the upper stratosphere for each additional K of temperature change. For kernels built from underlying atmospheric profiles where the top of the profile is not sufficiently high or with too coarse a resolution in the stratosphere, this additional upper stratospheric cooling is missed. In Fig. 3, it can be seen that the kernels based on a high-top atmospheric model with a high number of native model levels—ECHAM6, ECMWF-Oslo, ECMWF-RRTM and HadGEM3-GA7.1—have a marked increase in both the magnitude and the rate of negative LW outgoing flux at the 5 hPa and 1 hPa levels.

The consequences for a greenhouse-gas-induced stratospheric cooling are such that the additional stratospheric adjustment from greater cooling high in the stratosphere is not accounted for with either kernels or models that are truncated too low. Figure 4 shows the atmospheric temperature anomalies simulated in atmosphere-only simulations from CMIP6 models participating in RFMIP-ERF Tier 1 experiments (Pincus et al., 2016) for a 30-year time slice simulation where $CO_2$ concentrations are quadrupled relative to a pre-industrial control (piClim-4xCO2). Stratospheric cooling continues to increase above 10 hPa

**Table 1.** Radiative kernels considered in this study.

| Base model | Native model vertical levels | Top level (hPa) | 3rd level (hPa) | Reference |
|---|---|---|---|---|
| BMRC | 17 | 8.75 | 53.63 | Soden et al. (2008) |
| CCSM4 | 17 | 10 | 30 | Shell et al. (2008) |
| CESM | 30 | 3.64 | 14.36 | Pendergrass et al. (2018) |
| ECHAM5 | 19 | 10 | 50.39 | Previdi (2010) |
| ECHAM6 | 47 | 0.0099 | 0.11 | Block and Mauritsen (2013) |
| ECMWF-RRTMG | 24 | 1 | 6 | Huang et al. (2017) |
| ECMWF-Oslo | 60 | 0.11 | 0.5 | Myhre et al. (2018) |
| GFDL | 25 | 3.32 | 53.63 | Soden et al. (2008) |
| HadGEM2 | 38 | 2.99 | 13.02 | Smith et al. (2018) |
| HadGEM3-GA7.1 | 85 | 0.005 | 0.03 | this study |

in all models where data is available. A similar pattern of stratospheric cooling occurs in the piClim-ghg experiment which evaluates forcing from present-day greenhouse gases (not shown). Standard CMIP6 diagnostics call for model output on 19 pressure levels: 1000, 925, 850, 700, 600, 500, 400, 300, 250, 200, 150, 100, 70, 50, 30, 20, 10, 5 and 1 hPa, whereas in CMIP5 the standard set of 17 pressure levels did not include 5 and 1 hPa. Therefore, CMIP5 models were missing important additional stratospheric cooling where kernels were used for adjustment calculations.

The truncation of stratospheric height in "low top" radiative kernels (those with a top level lower in altitude than 1 hPa) has substantial consequences for adjustments to a greenhouse gas forcing. Figure 5 shows the stratospheric temperature adjustment for $4\times CO_2$ in 13 models contributing to RFMIP. A simplified tropopause definition is used here, borrowed from Soden et al. (2008), of a linear in latitude ramp from 100 hPa at the equator to 300 hPa at the poles. There is a spread of around 1 W m$^{-2}$ in calculated stratospheric temperature adjustment for each model using the full range of kernels, which is about 13% of the effective radiative forcing (ERF) for a quadrupling of $CO_2$ from these models (Smith et al., 2020). It can be seen in Fig. 5 that the kernel estimates for stratospheric adjustment to $CO_2$ forcing are clustered into two groups and one high outlier for most models. The one model where kernel estimates are not clearly separated into high and low clusters is GFDL-CM4 model, for which data is missing at 1 hPa. The "low-top" radiative kernels, with the exception of GFDL and ECHAM5, produce substantially lower estimates of the stratospheric temperature adjustment than the 'high-top' kernels (HadGEM3-GA7.1, ECHAM6 and ECMWF-Oslo). The high outlier, ECMWF-RRTM, has a large flux change at the 1 hPa level. We show in Section 5 that adjustments calculated using the HadGEM3-GA7.1 kernel in the IPSL-CM6A-LR model for a quadrupled $CO_2$ experiment provide small residuals (i.e. the adjustments are appropriately captured), suggesting that assuming there are no compensating errors, low-top kernels would underestimate the stratospheric temperature response and produce larger residuals.

To investigate the source of the differences between kernels in more detail, Fig. 6 shows the vertically integrated kernels from 1 hPa, 10 hPa, the full stratosphere (Soden et al. (2008) definition) and the full atmosphere. As previously, kernels that do

not include the uppermost layers have been extended based on the highest layer for which data is reported. Figure 6a gives the temperature adjustment from a uniform 1 K increase throughout the atmosphere. Here, there is little variation between kernels, and much of the total adjustment comes from the troposphere where the bulk of the atmosphere is present. Notably, the whole atmosphere adjustment to a 1 K temperature increase is of the order $-3$ W m$^{-2}$, approximating (if slightly underestimating) the Planck feedback ($-3.2$ W m$^{-2}$ K$^{-1}$).

Figure 6b shows the layer contributions when each kernel is applied to the atmospheric profile from the IPSL-CM6A-LR model for piClim-4xCO2. In contrast to the isothermal case, the top 1 hPa, top 10 hPa and stratospheric temperature adjustments show substantial diversity between kernels. The large contributions from these layers follow from Fig. 4 where the 1 and 10 hPa layers are of the order 10 K and 20 K cooler in the $4\times CO_2$ run than in the control. The HadGEM3-GA7.1, GFDL, ECMWF-Oslo, ECMWF-RRTM and ECHAM6 kernels have a larger adjustment from the top 10 hPa than the other kernels. Despite being nominally a "low-top" kernel, the GFDL kernel has a similar magnitude and gradient of cooling between 10 and 5 hPa as the high-top kernels (Fig. 3) and produces a similar result of total stratospheric adjustment to HadGEM3-GA7.1. The ECHAM5 kernel has more cooling around the 100 hPa level than any other kernel (Fig. 3) and also has stratospheric adjustments that are similar to HadGEM3-GA7.1.

The differences between models for the top 10 hPa propagate through to the tropopause level and whole atmosphere. Differences in tropospheric adjustments between kernels (whole atmosphere minus stratosphere) are small, showing that choice of atmospheric base state and radiative transfer is not critical for tropospheric temperature adjustments. This analysis gives further confidence that the choice of radiative kernel is not that important in climate feedback studies (assuming state changes are sufficiently small to remain linear; (Jonko et al., 2012)) as stratospheric temperature differences are small when differencing coupled atmosphere-ocean simulations (Chung and Soden, 2015b) and kernels show similar behaviour in the troposphere.

## 5   Linearity of the HadGEM3-GA7.1 kernel

This section shows the linear behaviour of the HadGEM3-GA7.1 kernel used with IPSL-CM6A-LR climate model output. IPSL-CM6A-LR is chosen as all required diagnostics are available in this model to evaluate linearity, including estimates of the IRF from double calls, and ISCCP simulator diagnostics for clouds. This model is also representative of the RFMIP population, with $4\times CO_2$ ERF and adjustments close to the multi-model average (Smith et al., 2020).

Linearity is evaluated by the size of the residual term, $\epsilon$, which is any TOA flux changes not explained by instantaneous radiative forcing or kernel-calculated adjustments. A guideline of linearity for the kernel method is that the residual should be within 10% of the ERF. We take two different approaches to calculate the residual. The first assumes that we have knowledge of the cloud adjustment term $A_c$ (e.g. from the ISCCP simulator or NASA A-Train kernels convoluted with ISSCP simulated cloud output from climate models (Zelinka et al., 2012; Yue et al., 2016)), and knowledge of the IRF ($F_i$) from a double-call in the online model. This is rare in practice, as double-calls and ISCCP cloud diagnostics are not routinely archived on the Earth System Grid Federation (ESGF), the main distributed data source for CMIP6 output (although model participation is

**Table 2.** IPSL-CM6A-LR double call results for $4\times CO_2$ experiments. IRFs are given in W m$^{-2}$.

| Base climatology | Second call | IRF LW | IRF SW | IRF Net | IRF LW CS | IRF SW CS | IRF Net CS |
|---|---|---|---|---|---|---|---|
| pre-industrial | $4\times CO_2$ | 3.66 | 0.83 | 4.49 | 5.02 | 0.46 | 5.48 |
| $4\times CO_2$ | pre-industrial | 4.94 | 0.81 | 5.75 | 6.26 | 0.46 | 6.72 |
| Mean | Mean | 4.30 | 0.82 | 5.12 | 5.64 | 0.46 | 6.10 |

improving for ISCCP clouds). From this "perfect information" method we can calcaulate an all-sky residual $\epsilon_{\text{all}}$ as

$$\epsilon_{\text{all}} = F - F_i - A_{T_a} - A_{T_s} - A_q - A_\alpha - A_c. \tag{6}$$

The second method does not require knowledge of the cloud adjustments, but does require knowledge of the clear-sky IRF, and in this case the clear-sky residual term $\epsilon_{\text{clr}}$ can be calculated as

$$\epsilon_{\text{clr}} = F^{\text{clr}} - F_i^{\text{clr}} - A_{T_a}^{\text{clr}} - A_{T_s}^{\text{clr}} - A_q^{\text{clr}} - A_\alpha^{\text{clr}}. \tag{7}$$

This clear-sky residual definition is more common in the literature (Smith et al., 2018; Soden et al., 2008; Vial et al., 2013), although often the clear-sky IRF is estimated rather than calculated directly. However, in some circumstances, clear-sky and all-sky IRF are known to be identically zero (e.g. in the LW spectrum to a change in the solar constant; Smith et al. (2018)). In these cases, Eq. (5) can be used with $F_i = F_i^{\text{clr}} = 0$ to determine $A_c$ which is then plugged into Eq. (6) to calculate $\epsilon_{\text{all}}$.

IPSL-CM6A-LR used two sets of double calls. In the RFMIP piClim-4xCO2 experiment (30-year time-slice atmosphere-only run with quadrupled $CO_2$) the second radiation call saw a pre-industrial $CO_2$ concentration. In the piClim-control experiment (pre-industrial atmosphere only run) the second radiation call saw $4\times CO_2$. The resulting IRF depends on the direction of the double call and is related to the underlying $4\times CO_2$ or pre-industrial climatology. The $4\times CO_2$ climatology that sees a pre-industrial second radiation call results in an IRF that is 1.26 W m$^{-2}$ greater than the pre-industrial climatology that sees a

$4\times CO_2$ second radiation call (Table 2). We take the mean of the two simulations to the be the IRF.

Table 3 shows ERF, IRF, adjustments and residuals using the HadGEM3-GA7.1 radiative kernel with the IPSL-CM6A-LR model output and cloud adjustments from the ISCCP simulator kernel. We use ISCCP kernel values in the calculation of $\epsilon_{\text{all}}$ and also compare the ISCCP values to the cloud-masking estimate of cloud adjustment from Eq. (5). For LW forcing the residuals are 0.28 W m$^{-2}$ for $\epsilon_{\text{clr}}$ and 0.38 W m$^{-2}$ for $\epsilon_{\text{all}}$. Residuals are present possibly due to a slight breakdown in the

linearity assumption for a forcing as large as $4\times CO_2$ (Jonko et al., 2012), however, the residuals are comfortably within the 10% linearity guideline. SW residuals are also within 10% of the ERF, with $\epsilon_{\text{clr}}$ being particularly small. For the net fluxes, forcings add but residuals partly cancel, such that $\epsilon_{\text{clr}}$ and $\epsilon_{\text{all}}$ are 3.2% and 1.9% of the ERF respectively.

Our results in Table 3 can be compared with the results of Zhang and Huang (2014) for 11 CMIP5 models. The instantaneous forcing and tropospheric adjustments from IPSL-CM6A-LR with the HadGEM3-GA7.1 kernel are in line with the CMIP5

forcing and adjustments except for the stratospheric temperature adjustment, which is outside the $2\sigma$ range from CMIP5 models (Table 4). As discussed in Smith et al. (2020), the $4\times CO_2$ ERF in available CMIP6 models (7.98 W m$^{-2}$) is (non-significantly)

**Table 3.** IPSL-CM6A-LR forcing and adjustments for the $4\times CO_2$ experiment using the HadGEM3-GA7.1 kernel. Fluxes are given in W $m^{-2}$. $A_{T_a}$ strat. and $A_{T_a}$ trop. are stratospheric and tropospheric temperature adjustments.

|  | ERF | IRF | $A_{T_a}$ strat. | $A_{T_a}$ trop. | $A_{T_s}$ | $A_q$ | $A_\alpha$ | $A_c$ (Eq. (5)) | $A_c$ (ISCCP kernel) | $\epsilon_{\mathrm{clr}}$ | $\epsilon_{\mathrm{all}}$ |
|---|---|---|---|---|---|---|---|---|---|---|---|
| LW | 5.33 | 4.31 | 2.74 | −1.38 | −0.49 | 0.52 |  | −0.66 | −0.75 | 0.28 | 0.38 |
| SW | 2.68 | 0.82 |  |  |  | 0.11 | 0.18 | 1.60 | 1.81 | −0.02 | −0.23 |
| Net | 8.01 | 5.12 | 2.74 | −1.38 | −0.49 | 0.63 | 0.18 | 0.94 | 1.06 | 0.26 | 0.15 |

**Table 4.** IPSL-CM6A-LR forcing and adjustments for the $4\times CO_2$ experiment using the HadGEM3-GA7.1 kernel and ISCCP cloud kernel (Zelinka et al., 2012) compared to the multi-model mean and standard deviation ($1\sigma$) in Zhang and Huang (2014) from 11 CMIP5 models. Fluxes are given in W $m^{-2}$. *Starred values are outside the $2\sigma$ range from Zhang and Huang (2014).

| Forcing or adjustment | IPSL-CM6A-LR | Zhang and Huang (2014) |
|---|---|---|
| ERF | 8.01 | 7.18 ($\pm$ 0.72) |
| IRF | 5.12 | 5.41 ($\pm$ 0.46) |
| Stratospheric temperature | *2.74 | 1.86 ($\pm$ 0.36) |
| Tropospheric + surface temperature | −1.87 | −1.66 ($\pm$ 0.21) |
| Water vapour (LW) | 0.63 | 0.42 ($\pm$ 0.12) |
| Clouds (LW) | −0.75 | −0.40 ($\pm$ 0.50) |
| Total SW | 1.89 | 1.55 ($\pm$ 0.83) |

greater than in corresponding CMIP5 sstClim4xCO2 experiments (7.53 W $m^{-2}$), which is also the case for IPSL-CM6A-LR in our comparison to Zhang and Huang (2014). IPSL-CM6A-LR is near the centre of the CMIP6 range for stratospheric temperature adjustment (Smith et al., 2020) so is a representative model of this ensemble. This could suggest stratospheric temperature adjustment increase as one driver of the increase in ERF between CMIP5 and CMIP6 models, although as most CMIP5 output is only on 17 model levels up to 10 hPa a formal comparison is difficult.

The stratospheric temperature adjustment is the only adjustment estimate that varies significantly between radiative kernels (Smith et al., 2018). If a kernel that did not resolve stratospheric temperature adjustment well was used instead of HadGEM3-GA7.1, this adjustment ($A_{T_a}$ strat. in Table 3) would be smaller, and the overall residuals for LW and net responses to $4\times CO_2$ larger. From Fig. 5 it can be seen that some kernels produce a stratospheric temperature adjustment around 0.7 W $m^{-2}$ lower than the HadGEM3-GA7.1 kernel, which would lead to residuals of the order 1 W $m^{-2}$ using these kernels, or more than 10% of the ERF.

## 6 Conclusions

This paper serves two purposes—it introduces the radiative kernel based on the high-top HadGEM3-GA7.1 general circulation model, and it compares estimates of the the stratospheric temperature adjustment obtained with a variety of different radiative kernels for quadrupled $CO_2$ experiments. The HadGEM3-GA7.1 kernel is the first to our knowledge that has been produced using a CMIP6 era model, with a focus on the 19 pressure level diagnostics available in CMIP6 output, although other kernels in the literature have used high-top atmospheric profiles (Huang et al., 2017; Myhre et al., 2018; Block and Mauritsen, 2013). Radiative kernels are produced for both top-of-atmosphere and surface fluxes and are available on the native 85-level hybrid height grid in addition to the 19 CMIP6 pressure levels.

We show that there is a significant diversity, of about 1 W m$^{-2}$ or 13% of the ERF for a quadrupling of $CO_2$, for estimates of stratospheric temperature adjustments to $CO_2$ depending on the radiative kernel used to derive the estimate. As tropospheric and land surface adjustments vary little between kernels to a variety of different forcing agents (Smith et al., 2018, 2020), these differences in stratospheric temperature adjustments lead to differing estimates of the total adjustment, and also of the IRF if it is calculated as a residual (Chung and Soden, 2015b, a; Soden et al., 2018). Climate feedbacks are little affected by the choice of kernel, due to the fact that stratospheric temperatures readjust quickly to an imposed forcing in coupled model simulations (Chung and Soden, 2015b).

While only one model (IPSL-CM6A-LR) archived IRF from a double call and a rigorous multi-model test is not possible, we show that the HadGEM3-GA7.1 kernel diagnoses IRF and adjustments with a small residual owing to the increased stratospheric resolution available compared to many CMIP3- and CMIP5-era kernels. We suggest that radiative kernels with a higher stratospheric resolution and model top are better able to fully capture stratospheric adjustments to $CO_2$ forcing in general, and generate smaller residuals. This effect has become more prominent with the additional 5 hPa and 1 hPa model levels archived as standard in processed CMIP6 model output compared to CMIP5. Archiving instantaneous radiative forcing from more models would be beneficial to further test the linearity assumption of the radiative kernel method.

*Data availability.* The HadGEM3-GA7.1 radiative kernels are available at https://doi.org/10.5281/zenodo.3594673 (Smith, 2019).

*Author contributions.* C.J.S. produced the HadGEM3-GA7.1 radiative kernel and led the writing of the manuscript. R.J.K. provided calculations of stratospheric adjustment to 4×$CO_2$ for all kernels considered in this paper. A.S. provided double call results from the IPSL-CM6A-LR model.

*Competing interests.* The authors declare no competing interests.

*Acknowledgements.* C.J.S. was supported by a NERC/IIASA Collaborative Research Fellowship (NE/T009381/1) and the European Union's Horizon 2020 research and innovation programme under grant agreement No 820829 (CONSTRAIN project). R.J.K. is supported by an appointment to the NASA Postdoctoral Program at NASA Goddard Space Flight Center. This work used the ARCHER UK National Super-computing Service (http://www.archer.ac.uk).

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

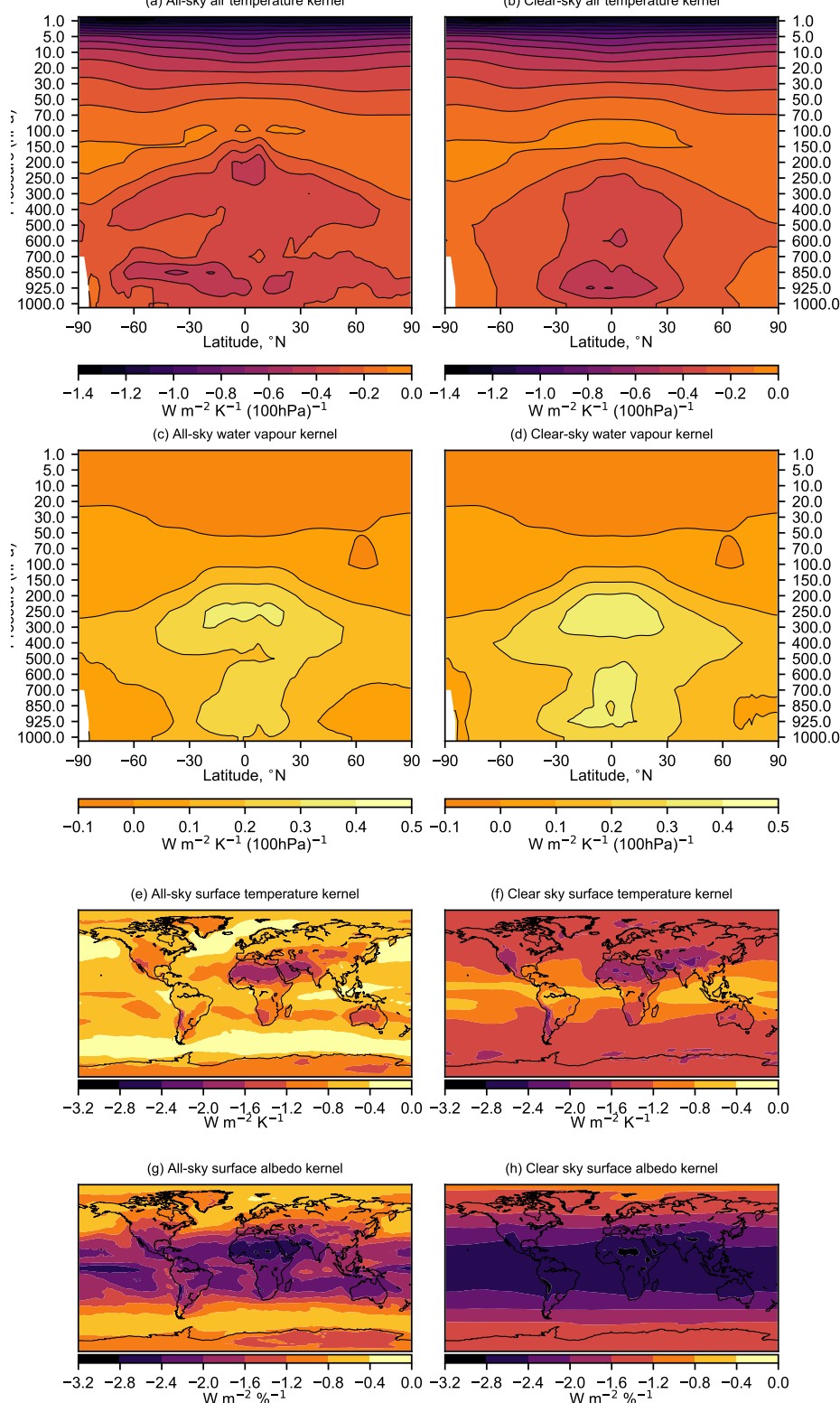

**Figure 1.** Top-of-atmosphere radiative kernels from HadGEM3-GA7.1.

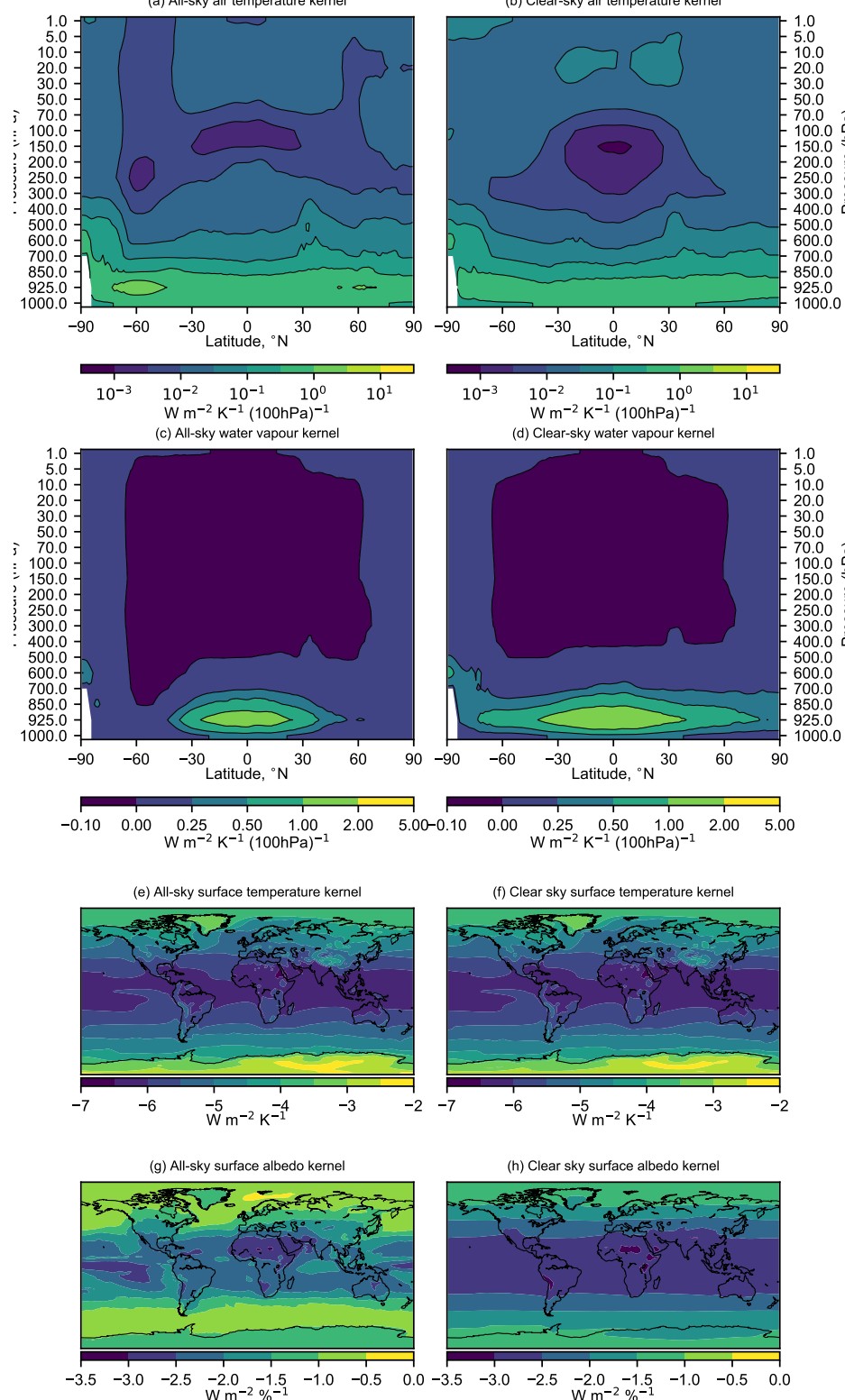

**Figure 2.** Surface radiative kernels from HadGEM3-GA7.1.

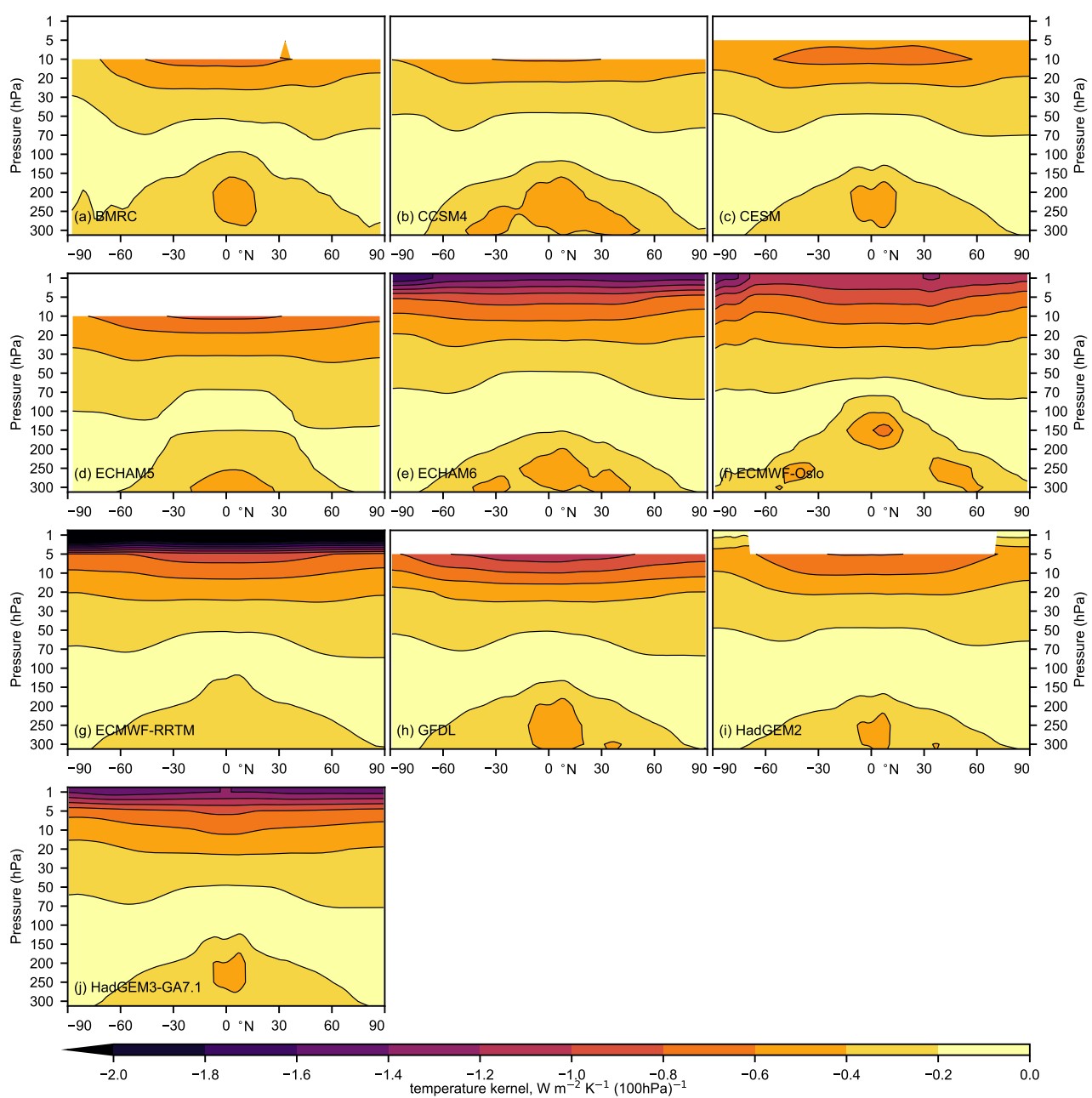

**Figure 3.** Air temperature radiative kernels available in the literature, truncated at 300 hPa to show the stratospheric temperature contribution. Blank areas are above the top level of the kernel.

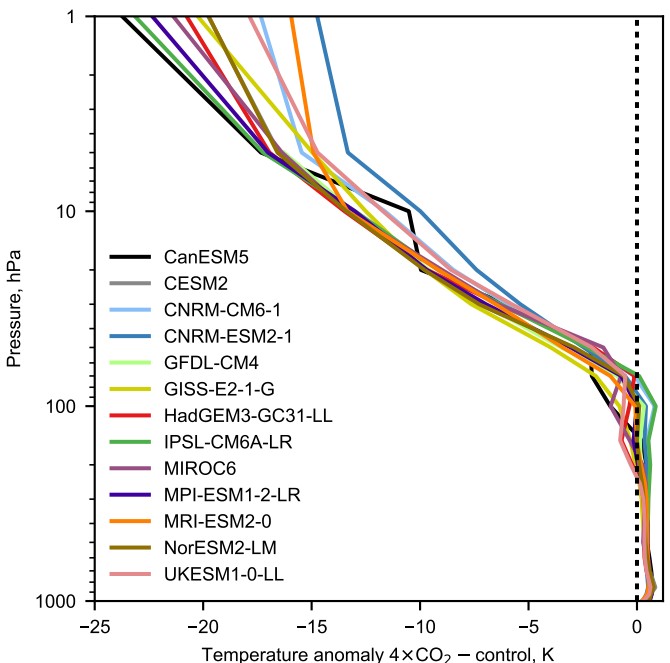

**Figure 4.** Atmospheric temperature differences for RFMIP models for the piClim-4xCO2 experiment minus piClim-control.

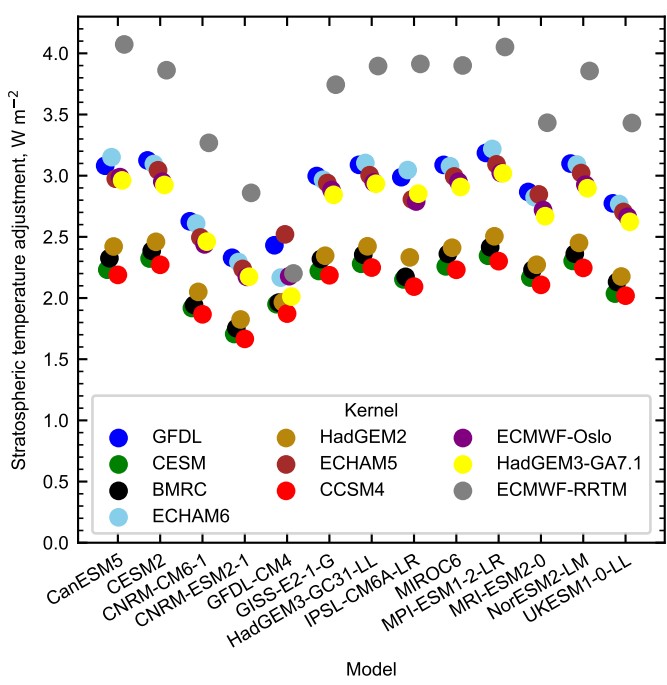

**Figure 5.** Stratospheric temperature adjustments calculated from RFMIP piClim-4xCO2 experiments using all kernels available in this study.

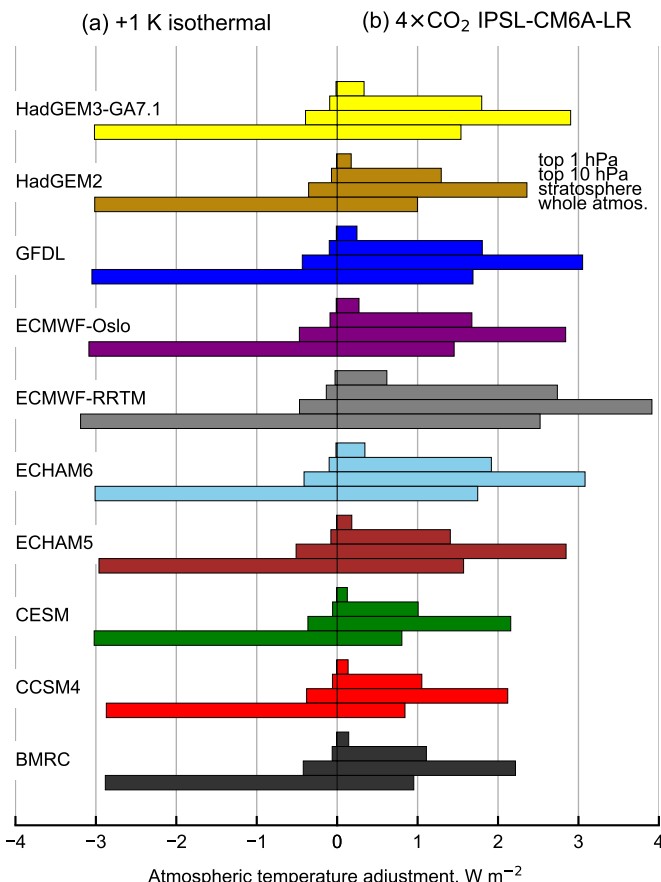

**Figure 6.** Layer contributions to the TOA temperature adjustment in each kernel considered in this study. From top to bottom the four bars show 1 hPa to TOA, 10 hPa to TOA, tropopause to TOA (i.e. stratospheric adjustment) and surface to TOA. (a) The effect of a uniform 1 K increase in atmospheric temperature on TOA fluxes, as diagnosed directly from the radiative kernel. (b) The kernels convoluted with atmospheric profiles from the IPSL-CM6A-LR model under an atmosphere-only quadrupled $CO_2$ run (RFMIP piClim-4xCO2 minus piClim-control).