# Peer review of "The HadGEM3-GA7.1 radiative kernel: the importance of a well-resolved stratosphere"

_Earth System Science Data, 2019_

## Referee Comment (RC1) · Yi Huang (Referee) · 1 Apr 2020

This paper has two objectives: 1) to present a new set of radiation kernels with high top, and 2) to intercompare this and a few other sets of radiation kernels, specifically with respect to the estimate of the radiative impact of stratospheric temperature adjustment in response to CO2. These are both potentially important contributions to make and warrant the efforts here. While I find the first objective well done and generally welcome a new kernel set to enrich the feedback analysis toolbox, I find the second objective relatively poorly executed. I'd suggest the authors take into consideration of the following comments and questions in revising this paper.

There lacks a solid basis for the "recommendation" of the three kernel datasets con-

cluded by the paper. For making such an important and strong statement as to which kernels are better, a principle (criterion) needs to be explicitly stated and justified for comparing them – this is currently missing in the paper. Note that a high-top kernel does not guarantee a higher accuracy in its assessment of radiative impact because the atmosphere which the kernel is based on can be biased – one should be especially cautious if the atmosphere is from a GCM – or because the radiation code used for the kernel computation is biased against the radiation code used in the target GCM simulation. On the other hand, a lower-top kernel also does not necessary lead to a poorer assessment, as shown by the GFDL kernel included here, due to fortuitous compensation of errors or due to some technical details of radiative transfer. Fore instance, some kernels may have used high-top atmospheric profiles in their computation but then truncated to lower top when applied to computing feedback. Moreover, computing and applying kernels at lower vertical resolution may be less subject to the nonlinear coupling between different vertical layers – one can test the non-radiation closure due to this issue, for example, by comparing the sum of vertical kernels to the true radiation change computed using the same radiation model from a vertically uniform 1-K temperature change.

To make a more objective and informative assessment, I suggest adding: 1) the comparisons of a) the global mean radiation change (Ax) due to layers above 1hPa, 10 hPa, tropopause and surface (whole column), respectively, assuming a uniform 1 K change of atmospheric temperature - this would disclose how the different kernels differ with respect to the radiative sensitivity to different portions of the atmosphere and whether there may be compensation of errors from different vertical portions; and b), like a), but using the atmospheric temperature adjustment to CO2 forcing as simulated by one representative GCM or the multiple model mean.

2) additional kernels, especially those observation-based kernels, such as the kernels of Huang et al. (2017) based on ERA-interim and of Yue et al. (2016) based on satellite. The former one (available from https://huanggroup.wordpress.com/research/) was

computed with a high-top atmospheric profile using RRTMG and provides kernel values up to 1hPa, which would provide a good comparison to ECMWF kernel here based on another radiation model (Oslo) – e.g., for assessing radiation code dependency noted above.

Additional comments: Line 17. It is recommended to include Zhang & Huang (2014) here, as this is one of the earliest that quantified $CO_2$ forcing, including both instantaneous forcing and the adjustment components, using kernels. The quantification of adjustment in multiple models reported by this work would make good comparisons to the results reported here, e.g., Table 2, 3.

Line 108. It is not obvious to me that cloud masking has a lesser impact on the surface flux. Please clarify and be more quantitative here.

Line 148-150. Can you illustrate the biases of these low-top kernels mentioned here?

Line 201. Again, the first that applied this residual method was Zhang&Huang [2014].

Line 10, 206. Can't approve such a "recommendation" for the reasons above. And such a recommendation could lead to wrongly denial of the use of the other kernels - both the lower top ones like the GFDL one that can achieve similar quantitative results and those the authors failed to include for comparison here.

References Huang, Y., Y. Xia and X. Tan, (2017), On the pattern of $CO_2$ radiative forcing and poleward energy transport, J. Geophys. Res.-Atmosphere, 122, 10,578–10,593, doi: 10.1002/2017JD027221. Kernel data: https://huanggroup.wordpress.com/research/

Yue, Q., Kahn, B. H., Fetzer, E. J., Schreier, M., Wong, S., Chen, X., & Huang, X. (2016). Observationâ $\breve{A}\check{R}$ based longwave cloud radiative kernels derived from the Aâ $\breve{A}\check{R}$ Train. Journal of Climate, 29(6), 2023–2040. https://doi.org/10.1175/JCLIâ $\breve{A}\check{R}$ Dâ $\breve{A}\check{R}$ 15â $\breve{A}\check{R}$ 0257.1

Zhang, M. and Y. Huang (2014), Radiative forcing of quadrupling CO2, J. of Climate,

27, 2496–2508. doi: http://dx.doi.org/10.1175/JCLI-D-13-00535.1

---

## Referee Comment (RC2) · Anonymous Referee #2 · 19 May 2020

This manuscript documents a new set of "radiative kernels," calculations of the sensitivity of radiative flux to atmospheric state, developed from a current-generation climate model with a domain reaching well into the stratosphere. The construction of the kernels was motivated by a desire to understand the fast response of stratospheric temperature to changes in carbon dioxide concentration and the authors demonstrate the value added by the new kernels. The construction of the kernels is described and their accuracy and generality assessed.

The data are well worth publishing. They require substantial computational resources to produce, extend the vertical domain in an almost-unique way, and use an accurate radiative transfer code. The free availability of the data has been verified. The manuscript is effective at documenting how the kernels are produced, providing enough

details for readers to understand and potentially replicate the steps. It is also effective at motivating why this implementation is useful, noting that the diagnosis of the fast climate response (the adjustment) to increased concentrations of carbon dioxide depends importantly on having a deep vertical domain. Beyond a few small points of expression noted below the manuscript could be most improved by more context for the uninitiated and a more general treatment of some ideas.

The introduction, which introduces the concept of and motivation for a radiative kernel, may be more general than is needed for the present manuscript. The generality makes it open to objections as to how the ideas are expressed. The general idea of a kernel is the ability to compute flux perturbations from state perturbations. As originally implemented by Soden, Shell, and others, these were restricted to specific characteristics of atmospheric state (air and surface temperature, water vapor, surface albedo, and excluding clouds) based partly on prioritization and partly on based on the availability of data. The does not establish a "standard" (line 47) nor does it exclude in principle other variables from being relevant (line 28). Readers may also wonder how the general material on the use of kernels (lines 39-54) is directly relevant to the construction of the present kernels.

The authors might revisit the introduction and focus it more tightly on the subject of the manuscript. This might include a not too profound explanation of how kernels can be used to diagnose both feedbacks and adjustments, and and explanation as to why yet another set of kernels might be desirable (i.e. the material that begins section 4). Care should be taken not to confuse routine practice with standardization.

Section 2:

It would be worth noting explicitly that these kernels rely on two almost distinct aspect of HadGEM: the radiation code SOCRATES run at low spectral resolution, and the climatology of atmospheric state including clouds, even if experience shows relatively weak dependence on the background state.

The authors might also explain some of their choices and any expected impacts. These might include the choice to develop kernels for pre-industrial conditions, the relatively highly-resolved vertical structure and coarse horizontal resolution of the simulations, and the high time resolution.

Section 4 illustrates the added value of the new kernels quite nicely. The use case is important but a little narrow. Is the value also added for other greenhouse gas forcings?

Section 5 is the least organized and clear and the section seems to assume a lot of background knowledge. The point of the section is to demonstrate the accuracy and applicability of the kernels. The narrative should be constructed to as to make this goal clear, explain how accuracy and applicability can be assessed, and finally to demonstrate the results.

Lines 2-3: "the utility of radiative kernels. . . is most appropriate" The last word isn't quite right. Utility can be greater or less but not appropriate.

Line 23: Kernels represent derivatives of flux with respect to state, not differential equations

Line 32: climate model (not mode)

Line 84: the equation should have units

Line 113: The sudden appearance of PDRMIP may confuse the uninitiated

Line 125: the limitations of low-topped kernels are presumably independent of whether the state comes from a "climate model" or any other source

Line 161: cars break down - what is meant here is "decomposition" or similar

Line 163: "ways of calculating the residual can be obtained" is a confusing phrasing.

---

## Author Comment (AC1) · 15 Jun 2020

This paper has two objectives: 1) to present a new set of radiation kernels with high top, and 2) to intercompare this and a few other sets of radiation kernels, specifically with respect to the estimate of the radiative impact of stratospheric temperature adjustment in response to CO2. These are both potentially important contributions to make and warrant the efforts here. While I find the first objective well done and generally welcome a new kernel set to enrich the feedback analysis toolbox, I find the second objective relatively poorly executed. I'd suggest the authors take into consideration of the following comments and questions in revising this paper.

»» Yi, thank you for the considered review of our paper. The first point about the addition of a new kernel is well taken and we are pleased that this is a useful addition to the existing set of model kernels. We note the deficiencies highlighted in your second point and trust we have addressed these satisfactorily in the response to your comments and in the forthcoming revised paper.

There lacks a solid basis for the "recommendation" of the three kernel datasets concluded by the paper. For making such an important and strong statement as to which kernels are better, a principle (criterion) needs to be explicitly stated and justified for comparing them – this is currently missing in the paper. Note that a high-top kernel does not guarantee a higher accuracy in its assessment of radiative impact because the atmosphere which the kernel is based on can be biased – one should be especially cautious if the atmosphere is from a GCM – or because the radiation code used for the kernel computation is biased against the radiation code used in the target GCM simulation. On the other hand, a lower-top kernel also does not necessary lead to a poorer assessment, as shown by the GFDL kernel included here, due to fortuitous compensation of errors or due to some technical details of radiative transfer. Fore instance, some kernels may have used high-top atmospheric profiles in their computation but then truncated to lower top when applied to computing feedback. Moreover, computing and applying kernels at lower vertical resolution may be less subject to the nonlinear coupling between different vertical layers – one can test the non-radiation closure due to this issue, for example, by comparing the sum of vertical kernels to the true radiation change computed using the same radiation model from a vertically uniform 1-K temperature change. To make a more objective and informative assessment, I suggest adding: 1) the comparisons of a) the global mean radiation change (Ax) due to layers above 1hPa, 10 hPa, tropopause and surface (whole column), respectively, assuming a uniform 1 K change of atmospheric temperature - this would disclose how the different kernels differ with respect to the radiative sensitivity to different portions of the atmosphere and whether there may be compensation of errors from different vertical portions; and b), like a), but using the atmospheric temperature adjustment to CO2 forcing as simulated by one representative GCM or the multiple model mean.

»» As described further on in the response the "recommendation" is weakened to a suggestion. We have included the comparisons that you suggest as additional figures.

2) additional kernels, especially those observation-based kernels, such as the kernels of Huang et al. (2017) based on ERA-interim and of Yue et al. (2016) based on satellite. The former one (available from https://huanggroup.wordpress.com/research/) was computed with a high-top atmospheric profile using RRTMG and provides kernel values up to 1hPa, which would provide a good comparison to ECMWF kernel here based on another radiation model (Oslo) – e.g., for assessing radiation code dependency noted above.

»» Thank you for the suggestion here. The Huang et al. (2017) kernels will be added to the analysis which were missed in the original submission. This kernel has a very strong negative temperature response at the 1 hPa level, in negative excess of -2 W m-2 (100hPa)-1 K-1 (shown in the update to figure 3) which provide large estimates of the stratospheric adjustment (update to figure 5).

»» The Yue et al. (2016) observational kernels focus on clouds and are more appropriately compared with the ISSCP simulator kernel (Zelinka et al., 2012) rather than our kernels derived from atmospheric state variables. While we use the ISCCP kernel to derive cloud radiative adjustments in the IPSL model from ISSCP simulator diagnostics, a comparison of cloud kernels produced by other groups is beyond the scope of this paper. But we thank you for making us aware of this paper and include references to it as further evidence of the utility of kernel approaches.

Additional comments: Line 17. It is recommended to include Zhang & Huang (2014) here, as this is one of the earliest that quantified CO2 forcing, including both instantaneous forcing and the adjustment components, using kernels. The quantification of adjustment in multiple models reported by this work would make good comparisons to the results reported here, e.g., Table 2, 3.

»» The reference to Zhang and Huang (2014) near line 17 for using kernels to diagnose

adjustments has been included, so thank you for reminding us of this study.

»» It is an excellent suggestion to compare our results to Zhang & Huang (2014). It in fact gives more weight to the claim that stratospheric changes are important. A new table 4 will be included comparing the results and showing that the IPSL-CM6A-LR model with our kernel is outside the 2-sigma range of the 11 models' stratospheric temperature adjustment in Zhang & Huang (2014). IPSL-CM6A-LR is fairly typical of CMIP6 models in terms of ERF and stratospheric adjustment as shown in Smith et al. (2020), and this paper also discusses the fact that ERF is increased in CMIP6 compared to CMIP5 for 4xCO2. Therefore we speculate that an increase in stratospheric temperature adjustment could be partially responsible for an increase in CMIP6 ERF, although we can't prove it without IRF calculations from more models and the fact that most CMIP5 model output does not include the 5 hPa and 1 hPa levels, again preventing a formal comparison.

Line 108. It is not obvious to me that cloud masking has a lesser impact on the surface flux. Please clarify and be more quantitative here.

»» In figures 1 and 2 the subfigures were actually mislabelled. This statement was meant to refer to (what is erroneously labelled) the difference between 2(c)-2(g) and 1(c)-1(g), which are the surface temperature kernels for all-sky and clear-sky for the surface and TOA. In the attachment to this review the differences are shown. The differences in the TOA kernels are large but for the surface kernels are small. The figure captions and surrounding text will be updated. Thank you for spotting this.

[Figure A: differences between surface temperature kernels for TOA and surface fluxes]

Line 148-150. Can you illustrate the biases of these low-top kernels mentioned here?

»» To keep the structure of the paper as it is we refer to the following section where we show the decomposition in the IPSL-CM6A-LR model using our kernel but explain the bias here. Using either equation 7 or 8 we get a small positive residual (table 3).

[Figure]

Figure 5 implies that the low-top kernels (excluding GFDL) are around 0.5 W m-2 or more lower in their stratospheric adjustment than HadGEM3-GA7.1, so that the residuals would be more positive assuming the tropospheric adjustments are similar across kernels. Although we don't compare non-stratospheric adjustments in this paper, I showed previously that kernels agree well for tropospheric adjustments (Smith et al., 2018), as they do for climate feedbacks (e.g. Soden et al., 2008).

»» The following addition to the manuscript is made around line 148:

»» "We show in section 5 that adjustments calculated using the HadGEM3-GA7.1 kernel in the IPSL-CM6A-LR model for a quadrupled CO2 experiment provide small residuals (i.e. the adjustments are appropriately captured), suggesting that assuming there are no compensating errors, low-top kernels would underestimate the stratospheric temperature response and produce larger residuals."

Line 201. Again, the first that applied this residual method was Zhang&Huang [2014].

»» Thank you for this suggestion. This reference has been added here.

Line 10, 206. Can't approve such a "recommendation" for the reasons above. And such a recommendation could lead to wrongly denial of the use of the other kernels - both the lower top ones like the GFDL one that can achieve similar quantitative results and those the authors failed to include for comparison here.

»» We are inclined to agree that for a data description paper recommendation may be a bit strong and have changed the sentence near the end:

»» "We suggest that radiative kernels with a higher stratospheric resolution and model top are better able to fully capture stratospheric adjustments to CO2 forcing in general, and generate smaller residuals. This effect has become more prominent with the additional 5 hPa and 1 hPa model levels archived as standard in processed CMIP6 model output compared to CMIP5."

»» and in the abstract:

»» "We show in the IPSL-CM6A-LR model where a full set of climate diagnostics are available that the HadGEM3-GA7.1 kernel exhibits linear behaviour and the residual error term is small, and from a survey of kernels available in the literature that in general low-top radiative kernels underestimate the stratospheric temperature response."

References

Huang, Y., Y. Xia and X. Tan, (2017), On the pattern of CO2 radiative forcing and poleward energy transport, J. Geophys. Res.- Atmosphere, 122, 10,578–10,593, doi: 10.1002/2017JD027221.

Kernel data: https://huanggroup.wordpress.com/research/

Yue, Q., Kahn, B. H., Fetzer, E. J., Schreier, M., Wong, S., Chen, X., & Huang, X. (2016). ObservationâAËŸ Rbased longwave cloud radiative ker- ËĞ nels derived from the AâAËŸ RTrain. Journal of Climate, 29(6), 2023–2040. ËĞ https://doi.org/10.1175/JCLIâAËŸ RDâ ËĞ AËŸ R15â ËĞ AËŸ R0257.1 ËĞ

Zhang, M. and Y. Huang (2014), Radiative forcing of quadrupling CO2, J. of Climate, 27, 2496–2508. doi: http://dx.doi.org/10.1175/JCLI-D-13-00535.1

(1e - 1f) All-sky minus clear-sky TOA surface temperature kernel

W m$^{-2}$ K$^{-1}$

(2e - 2f) All-sky minus clear-sky surface surface temperature kernel

W m$^{-2}$ %$^{-1}$

---

## Author Comment (AC2) · 15 Jun 2020

This manuscript documents a new set of "radiative kernels," calculations of the sensitivity of radiative flux to atmospheric state, developed from a current-generation climate model with a domain reaching well into the stratosphere. The construction of the kernels was motivated by a desire to understand the fast response of stratospheric temperature to changes in carbon dioxide concentration and the authors demonstrate the value added by the new kernels. The construction of the kernels is described and their accuracy and generality assessed.

The data are well worth publishing. They require substantial computational resources to produce, extend the vertical domain in an almost-unique way, and use an accurate radiative transfer code. The free availability of the data has been verified. The manuscript is effective at documenting how the kernels are produced, providing enough details for readers to understand and potentially replicate the steps. It is also effective at motivating why this implementation is useful, noting that the diagnosis of the fast climate response (the adjustment) to increased concentrations of carbon dioxide depends importantly on having a deep vertical domain. Beyond a few small points of expression noted below the manuscript could be most improved by more context for the uninitiated and a more general treatment of some ideas.

»» Thank you for your positive overall comments and we are pleased that you agree that the data and description paper is worth publishing in a form close to present.

The introduction, which introduces the concept of and motivation for a radiative kernel, may be more general than is needed for the present manuscript. The generality makes it open to objections as to how the ideas are expressed. The general idea of a kernel is the ability to compute flux perturbations from state perturbations. As originally implemented by Soden, Shell, and others, these were restricted to specific characteristics of atmospheric state (air and surface temperature, water vapor, surface albedo, and excluding clouds) based partly on prioritization and partly on based on the availability of data. The does not establish a "standard" (line 47) nor does it exclude in principle other variables from being relevant (line 28). Readers may also wonder how the general material on the use of kernels (lines 39-54) is directly relevant to the construction of the present kernels.

»» Thank you for these suggestions. In our opinion the introduction is not overly long at present so we are inclined to keep lines 39-54 in the paper as general background. We note that others have taken a different approach and have assumed more background knowledge (e.g. Prendergrass et al 2018). Readers familiar with radiative kernels could easily skip over the introduction. Those not familiar may welcome it, and as the kernel method is used increasingly outside the climate feedback community in which it was developed, it may make the paper more self-contained.

»» Line 47 has been changed to "Cloud adjustments and feedbacks cannot be determined directly using atmospheric state kernels". The previous wording could be taken to imply that cloud adjustments/feedbacks could not be calculated at all using only atmospheric kernels, which was not the intention.

»» Line 28: after this sentence, included "Although other (non-cloud) variables may also be relevant, the majority of adjustments are expected to be captured under this framework (Vial et al., 2013)."

The authors might revisit the introduction and focus it more tightly on the subject of the manuscript. This might include a not too profound explanation of how kernels can be used to diagnose both feedbacks and adjustments, and and explanation as to why yet another set of kernels might be desirable (i.e. the material that begins section 4). Care should be taken not to confuse routine practice with standardization.

»» To improve the motivation, the first paragraph from section 4 has been assimilated into the introduction. Following the previous response, we believe the present level of background is appropriate.

Section 2:

It would be worth noting explicitly that these kernels rely on two almost distinct aspect of HadGEM: the radiation code SOCRATES run at low spectral resolution, and the climatology of atmospheric state including clouds, even if experience shows relatively weak dependence on the background state.

This section has been updated to include a sentence:

»» "The kernel is therefore dependent on two aspects of the HadGEM3-GA7.1 model: the pre-industrial background climatology (including clouds), and the broadband version of the radiation code."

The authors might also explain some of their choices and any expected impacts. These might include the choice to develop kernels for pre-industrial conditions, the relatively

highly-resolved vertical structure and coarse horizontal resolution of the simulations, and the high time resolution.

»» In the updated section 2 we have added a few sentences at opportune points, justifying some of the methodological choices. More explanation for each is given below.

»» The pre-industrial conditions were chosen for the kernel base state as these kernels were designed to be used first and foremost with the RFMIP piClim-X forcing experiments (X = 4xCO2, present-day GHG, present-day aerosols, present-day land-use and present-day total anthropogenic), covering both negative and positive forcing relative to the pre-industrial. The base climate of the target models is pre-industrial except for the perturbed component(s) and differences are taken with respect to piClim-control (an atmosphere-only pre-industrial control run). Therefore, a pre-industrial climatology for the kernel is appropriate. Other choices were of course possible such as linearising around present-day conditions.

»» The relatively high (for a GCM) vertical resolution is the default configuration of the HadGEM3-GA7.1 model, and its importance for stratospheric adjustment is already stated.

»» The horizontal resolution (1.875° x 1.25°) is the lowest resolution used for HadGEM3 and UKESM1 in CMIP6. Many other CMIP6 models that are ultimately the target of the radiative kernels (excluding HighResMIP) are on similar resolutions to this. We are not aware of an exhaustive list, but Table 1 in Smith et al. (2020) lists those used in RFMIP, so it is not apparent that an increase in resolution in the base climate would improve accuracy after the kernels are re-gridded to the resolution of target climate models. But the main reason why a finer resolution wasn't considered is computing time. The "MM" resolution of HadGEM3 has an atmospheric grid of 0.83° x 0.56° with 5 times as many grid points, so running the offline radiation model would have taken 5 times longer. To do this in a reasonable amount of time would require

a reduction in the vertical or time resolution (or use of HPC). The high vertical resolution of these kernels are one of their strengths, and it would be difficult to reduce the number of temporal radiation calls as explained below.

»» A two-hour time step was used as instantaneous rather than time-mean climate output is needed to run the offline radiative transfer code (a discussion of why time-mean output does not work well is in Bellouin et al, 2020; section 4). Two hours was considered fine enough to avoid biases by undersampling the diurnal cycle of temperature, humidity and clouds, and shortwave solar geometry. On the last point, using longer timesteps like 6 hours between shortwave calls leads to some longitudes receiving substantially more incoming solar radiation than others over the course of the year. This effect could be achieved alternatively by using a timestep that does not divide 24 (e.g. 22, 23, 25 or 26 hours) and running several years of the climate model to sample diurnal, seasonal and interannual variability, such as is sometimes done in PRP calculations.

Section 4 illustrates the added value of the new kernels quite nicely. The use case is important but a little narrow. Is the value also added for other greenhouse gas forcings?

»» Thank you for your positive comments here and agree it is beneficial to demonstrate why the kernels are useful. In response to an earlier comment we moved the first paragraph to the introduction.

»» It is indeed useful for other GHG forcings. We generalise from CO2 to GHGs in a few places in Section 4, although the focus of the results will still be on CO2 as we have the double-call results from IPSL-CM6A-LR for this experiment.

»» We didn't re-run every kernel calculation using the piClim-ghg experiment in this paper. However, in supplementary figure 1 in Smith et al. (2020) we compare four of the kernels for all RFMIP experiments, showing that HadGEM3, ECMWF-Oslo and GFDL kernels show larger stratospheric adjustments for piClim-ghg than the CCSM kernel which is in accordance with the 4xCO2 results in this paper. In attachment to

this response we show the temperature profile for the GHG experiment, which moves in the same direction as for 4xCO2 but with a lower magnitude.

[Figure B: stratospheric temperature differences from piClim-ghg minus piClim-control for 13 models in RFMIP].

»» Also note that in figure 4 we have removed the missing top level from GFDL-CM4, which was previously displayed as zero rather than missing.

Section 5 is the least organized and clear and the section seems to assume a lot of background knowledge. The point of the section is to demonstrate the accuracy and applicability of the kernels. The narrative should be constructed to as to make this goal clear, explain how accuracy and applicability can be assessed, and finally to demonstrate the results.

»» Thank you for this suggestion. It has been re-written to introduce the aim of the exercise (the discussion on IRF appears too soon and is given too much weight) before showing the results. Following suggestions from reviewer #1 we have also included a comparison of the results here to a previous study from CMIP5. To address the assumption of background knowledge on interpretation of section 5, we note that the introduction section may come in useful here to the uninitiated reader, which is a reason why we feel it is appropriate to keep most of it.

Lines 2-3: "the utility of radiative kernels. . . is most appropriate" The last word isn't quite right. Utility can be greater or less but not appropriate.

»» Thanks for pointing out the confusing wording. Changed "most appropriate" to "greatest".

Line 23: Kernels represent derivatives of flux with respect to state, not differential equations

»» A slight terminological liberty taken on our part. We'll keep the notation of eq. (1) as it is used by others (e.g. Shell et al., (2008) where they have used (F-Q) in place

of R, and Huang et al. (2017)) and provides a nice concise representation but explain that it isn't strictly a differential equation.

Line 32: climate model (not mode)

»» Typo corrected - thank you

Line 84: the equation should have units

»» Updated to confirm that 10000 and p_thick are both in units of Pa here.

Line 113: The sudden appearance of PDRMIP may confuse the uninitiated

»» PDRMIP (Precipitation Driver and Response Model Intercomparison Project) acronym is now introduced, which was an oversight in the first submission.

Line 125: the limitations of low-topped kernels are presumably independent of whether the state comes from a "climate model" or any other source

»» Agreed: revised this sentence to be more general: "For kernels built from underlying atmospheric profiles where the top of the profile is not sufficiently high or with too coarse a resolution in the stratosphere, this additional upper stratospheric cooling is missed."

Line 161: cars break down - what is meant here is "decomposition" or similar

»» As part of the re-write of section 5, this sentence will be revised.

Line 163: "ways of calculating the residual can be obtained" is a confusing phrasing

»» Agree this is not meaningful: changed to "Two different ways of calculating the residual exist."

[Figure]

[Figure]

**Fig. 1.** Figure B: stratospheric temperature differences from piClim-ghg minus piClim-control for 13 models in RFMIP

---

## Author Response (AR2)

[revised manuscript text omitted]

Thank you David for the technical check. Responses below:

**Small questions / concerns:**

**1) Some confusion or uncertainty induced by the sentence at lines 74 to 76. Please revise if necessary.**

Hopefully breaking this sentence into two and describing two example use cases makes this clearer:

> A pre-industrial base climatology for the kernels was chosen because the first identified use case for this kernel set was the RFMIP-ERF Tier 1 single-forcing experiments (Pincus et al., 2016). These experiments compare positive (e.g. greenhouse gas) and negative (e.g. aerosol) forcing perturbations with respect to a pre-industrial control baseline.

**2) Line 109: I think Copernicus standards will insist on Fig rather than fig. These changes will happen at typesetting but you could speed the process by doing a search and replace now. Same comment for (line 130) Table vs table.**

We used the `cleveref` LaTeX package, and a simple option to capitalise the Fig. and Table references has corrected this. This is not highlighted in the marked up changes as the LaTeX body text is unchanged.

**3) Line 198: You introduce the ESGF acronym without defining Earth System Grid etc...**

Now fixed.

**4) Line 214: Small text errors here?**

I have tried to be more explicit and systematic in the description.

> The resulting IRF depends on the direction of the double call and is related to the underlying $4 \times CO_2$ or pre-industrial climatology. The $4 \times CO_2$ climatology that sees a pre-industrial second radiation call results in an IRF that is 1.26 W m$^{-2}$ greater than the pre-industrial climatology that sees a $4 \times CO_2$ second radiation call (Table 2).

**5) A quick search shows that you have neglected to define the ISCCP acronym?**

Agreed, now fixed.